# Critical Review of the Increasing Complexity of Access and Benefit-Sharing Policies of Genetic Resources for Genebank Curators and Plant Breeders–A Public and Private Sector Perspective

**DOI:** 10.3390/plants12162992

**Published:** 2023-08-19

**Authors:** Andreas W. Ebert, Johannes M. M. Engels, Roland Schafleitner, Theo van Hintum, Godfrey Mwila

**Affiliations:** 1Independent Researcher, 73529 Schwäbisch Gmünd, Germany; 2Independent Researcher, Voc. Podere Sansano 5, 06062 Citta’ della Pieve (PG), Italy; j.engels@cgiar.org; 3World Vegetable Center, 60 Yi-Min Liao, Shanhua, Tainan 74151, Taiwan; roland.schafleitner@worldveg.org; 4Centre for Genetic Resources, the Netherlands (CGN), Wageningen University & Research, 6700 AA Wageningen, The Netherlands; theo.vanhintum@wur.nl; 5Executive Secretary of the Zambia Seed Trade Association (ZASTA), Sulmach Buldings, Tiyende Pamodzi, Off Nangweya, Lusaka, Zambia; godfrey.mwila@gmail.com

**Keywords:** vegetable genetic resources, global germplasm conservation and use systems, plant breeding, access and benefit-sharing, digital sequence information, international treaty for plant genetic resources, Convention on Biological Diversity, Nagoya Protocol

## Abstract

Plant breeders develop competitive, high-yielding, resistant crop varieties that can cope with the challenges of biotic stresses and tolerate abiotic stresses, resulting in nutritious food for consumers worldwide. To achieve this, plant breeders need continuous and easy access to plant genetic resources (PGR) for trait screening, to generate new diversity that can be built into newly improved varieties. International agreements such as the Convention on Biological Diversity (CBD), the International Treaty on Plant Genetic Resources for Food and Agriculture (ITPGRFA) and the Nagoya Protocol recognised the sovereign rights of countries over their genetic resources. Under the CBD/Nagoya Protocol, countries are free to establish specific national legislations regulating germplasm access and benefit-sharing to be negotiated bilaterally. Consequently, access to PGR became increasingly restricted and cumbersome, resulting in a decrease in germplasm exchange. The ITPGRFA attempted to ease this situation by establishing a globally harmonised multilateral system (MLS). Unfortunately, the MLS is (still) restricted to a limited number of food and forage crops, with very few vegetable crops. Easy and continuous access to genetic diversity combined with equitable and fair sharing of derived benefits is a prerequisite to breeding new varieties. Facilitated access contributes to sustainable crop production and food and nutrition security; therefore, access to and, consequently, use of PGRFA needs to be improved. Thus, the authors recommend, among others, expanding the scope of the ITPGRFA to include all PGRFA and making them and all related information accessible under a Standard Material Transfer Agreement (SMTA) combined, if necessary, with a subscription system or a seed sales tax. Such a transparent, functional and efficient system would erase legal uncertainties and minimise transaction costs for conservers, curators and users of genetic resources, thus aiding plant breeders to fulfil their mission.

## 1. Introduction

Free access to and exchange of germplasm have been the foundation for all plant domestication and improvement efforts since the start of sedentary farming. Through most of human history, access has been constrained by physical distance and limited knowledge, not by an unwillingness to share or legal instruments. Until the signing of the Convention on Biological Diversity (CBD) in 1992–1993 (Table 1), germplasm was considered a common heritage of humankind to be preserved and to be freely available for use, for the benefit of present and future generations as per the International Undertaking (IU) established by the FAO Commission on PGR in 1983 [1,2,3]. Plant breeders obtained the required germplasm for their crop-improvement efforts from a wide variety of existing commercial varieties, public and private genebanks, public and private collecting missions, working collections maintained at research institutions and private companies, and from farmers’ fields and stores.

Plant breeding is a long and tedious process and requires a lot of investment. Vegetable seed companies use up to 30% of their turnover for research and development. With the aim of encouraging continuous development of new plant varieties for the benefit of society at large, plant breeders’ rights (PBR) were introduced through the creation of plant variety protection and internationally harmonised through the International Union for the Protection of New Varieties of Plants (UPOV) Convention, adopted in Paris in 1961 and revised in 1972, 1978 and 1991 [4]. Article 15 of the UPOV Convention provides a compulsory breeders’ exemption to the exclusive right [5], allowing everyone to freely use any protected variety for further breeding and commercialising the new ones without any obligation to the original PBR holder as long as the newly developed product is sufficiently different from the protected variety. This provision constitutes an essential and principal element towards ensuring continued access of plant breeders worldwide to elite privately owned germplasm as parental material [6].

With the advent of biotechnological innovations during the 1980s, some countries allowed certain inventions to be protected through patents. The patenting of biotechnological inventions can be traced back to 1980 when the Supreme Court of the United States decided that a genetically modified organism, in that specific case a bacterium, is patentable [7]. Thereafter, several proprietary products were released in plant sciences, such as traits/genes and genetically engineered varieties.

Irregular access and use of genetic resources and related traditional knowledge of countries, indigenous peoples and local communities without their consent and the patenting of derived or associated information for further commodification is understood as biopiracy [8]. Cases of biopiracy and the perception in the Global South that the breeding industry in the Global North was earning money based on the genetic resources collected in the Global South without sharing due benefits were major reasons why the continuous free availability and accessibility of genetic resources as foreseen under the IU was no longer considered an acceptable paradigm [9]. This led to the development of new global legal frameworks (see Section 3). 

Intergovernmental negotiations with the aim of protecting and conserving biological resources, making them available under the assumption of sharing benefits derived from their use on agreed terms, led to the adoption of several international agreements, such as the CBD in 1992 and the subsequent so-called Nagoya Protocol in 2010, that advise countries on how to implement Access and Benefit-sharing (ABS) regulations in their national legislations, and the International Treaty on Plant Genetic Resources for Food and Agriculture (ITPGRFA) in 2001 (Table 1). Section 3 of this paper will deal with these international agreements in more detail. Both the CBD and the ITPGRFA had, amongst others, the objective of facilitating access to PGRFA [2]). However, due to low compliance and complicated, rather bureaucratic implementation procedures, especially under the bilateral regulations of the Nagoya Protocol, it did not have the desired effect. 

Screening programs for desirable traits in crop breeding require access to a large quantity of plant genetic resources from different sources and countries. At the end of the screening process, only a few breeding lines will be used in generating the final, new commercial variety. Negotiating and securing PIC and MAT for each germplasm source via bilateral agreements is a complex and time-consuming process [10]. Differing national and even local ABS laws and regulations create a significant entry barrier and represent a major challenge for seed companies to establish a relevant collection of starting materials for breeding. Bilateral ABS contracts under the Nagoya Protocol require extensive tracking and tracing of every germplasm transfer and subsequent use and movement worldwide. Tracking systems are meant to provide the link between access and use by following the international movements of genetic resources, from original provision until the inclusion in a commercial product, either a new plant variety or other inventions, which could be patentable [11]. Commercial plant breeding programmes, in general, have multiple breeding cycles running in parallel, often in different countries with different climatic conditions and seasons, involving exchanges of breeding material between countries. In this process, ABS tracking requirements create significant complexity in the breeding workflow [10]. For the aforementioned reasons, access to germplasm is often limited under the Nagoya Protocol [10,12,13]. The unresolved regulation of access to digital sequence information (DSI) associated with germplasm accessions complicates things even further [14], resulting in a further decline in the use of genetic resources for crop improvement. 

In order to cope with the ever-increasing threats from biotic and abiotic stresses, exacerbated by climate change, the recommendation to consume more biodiverse food to counter the increase in diet-related diseases, and the need to feed a still-growing global population with healthy diets, plant breeders do need continuous access to cultivated and non-cultivated (crop wild relatives-CWR) genetic diversity for trait screening and to generate new diversity with the aim of developing competitive, high-yielding, and nutritious new varieties for the farming community. Loss of crop diversity and erosion of genetic diversity due to a variety of reasons is of general concern, requiring continued efforts to mitigate further loss by safeguarding crop diversity ex situ [15]. However, the decision process to obtain collecting permits and to acquire germplasm in compliance with recently developed legal requirements of international rules and regulations is complex and cumbersome and creates uncertainties for genebank curators and plant breeders alike [10,12,13]. Moreover, the complex/bureaucratic accessibility of genetic resources becomes an additional criterion for deciding whether or not to use genetic resources regulated by the CBD and the ITPGRFA. Although the International Treaty established standard rules and procedures (i.e., the SMTA) for accessing PGRFA, it does not prevent member countries from implementing related legislation, for example, concerning the inclusion of specific PGRFA in the MLS [13]. Such legislation might differ from country to country.

By its very nature, the ABS procedures under the CBD/Nagoya Protocol differ from country to country, are often unclear and highly bureaucratic, and are still evolving. In India, for example, the National Biodiversity Authority and the State Biodiversity Boards are required to consult with the respective local Biodiversity Management Committee (out of approx. 270,000 existing in the country) regarding the access conditions expected by the conservers and holders of biological resources and associated traditional knowledge. Once this internal consultation process is completed, foreign users need to “negotiate” the contractual ABS clauses with national authorities [10]. Public breeders and breeders from small and medium-sized enterprises do not usually have the necessary expertise and resources to navigate such complex arrangements and, therefore, often prefer to stay away from such complexity.

Mekonnen and Spielman [16] correlated historical trends in genebank acquisitions and changes in germplasm exchange over time, with changes in the international policy environment for seven crops that are essential for food security in developing countries. Based on these results, the authors concluded that a country’s membership in the CBD is closely associated with reductions in the flow of genetic resources and that the Nagoya Protocol may affect global PGRFA flows in a potentially negative and unintended manner. In contrast, ITPGRFA membership is likely to moderate the negative effects of the CBD and the Nagoya Protocol [16].

Nutritionists and health sector specialists are increasingly highlighting the role of vegetables, fruit and nuts for their potential in combating the triple burden of malnutrition (undernutrition, hidden hunger and overnutrition) [17]. Unfortunately, facilitated access to vegetable genetic resources under the less cumbersome multilateral agreement of the ITPGRFA is rather limited, as the majority of vegetable crops are not included in the Annex I list of the MLS and thus, automatically fall under the Nagoya Protocol obligations. However, genetic diversity is needed to develop new resilient varieties with multiple resistances against ever-increasing biotic stresses and tolerance to abiotic stresses that are exacerbated by climate change. Therefore, the authors of this review emphasise in particular the case of vegetable genetic resources due to their unique role in nutrition security. Vegetable breeders from the public and private sectors face considerable difficulties in accessing and using the required genetic diversity for breeding elite, nutrient-dense and resilient vegetable crop varieties. The vegetable breeding sector deals with a wide range of species and an enormous diversity of diseases and insect pests and is, therefore, perhaps even more reliant on germplasm from genebanks than breeders dealing with other horticultural and agronomic crops. Nevertheless, the challenges and legal uncertainties in accessing and using germplasm and related information for breeding discussed in this paper apply to all PGRFA, and most references cited are not restricted to vegetable crops.

This paper highlights the importance of genetic diversity and plant breeding for sustainable agricultural production and food and nutrition security. In this context, the authors focus on the increasing complexity of access and benefit-sharing policies and their implications for crop germplasm collecting and conservation, and access to and utilisation of the conserved crop genetic diversity by plant breeders from the public and private sectors. Several options for addressing current constraints regarding ABS of PGRFA are discussed. It is essential to develop a more satisfying and functional global germplasm conservation and use system to halt further genetic erosion of threatened and endangered PGRFA and preserve it for use by current and future generations of breeders, farmers and consumers, and society as a whole.

## 2. The Importance of Genetic Diversity and Plant Breeding for Agricultural Production and Food and Nutrition Security

Since the transition from hunting–gathering to sedentary farming, producing enough food for a growing population has always been a significant challenge. The origins of agriculture can be traced back to about 12,000 years ago, when wheat and barley domestication and cultivation started in the Fertile Crescent in the Near East [18], and a ‘crop package’ spread from there into Europe, Asia and Africa several thousand years later. Climate change and population growth are considered to have major impacts on sedentary farming. Today, population growth and greater per capita purchasing power, coupled with higher meat, dairy and egg consumption, and the use of agricultural crops for biofuel production are considered to be major driving forces for the continuously growing global demand for food, fibre and fuel crops until 2050 and beyond [19,20]. However, the increasing human population, scarcity of fertile land for the expansion of cropping areas, the negative impact of agriculture on the environment and the increasing threats from climate change mean that further increases in food production must primarily be based on yield enhancement and productivity growth. This can be achieved through continuous plant breeding efforts and sustainable intensification of crop production practises on existing croplands, on which current crop yields are well below the yield potential [21].

In many parts of the world, plant breeding has contributed considerably to increased productivity, apart from increased use of agricultural inputs such as irrigation water, chemical fertilisers and pesticides. This led to stable markets, lower food prices and reduced price volatility [22,23], among others, evidenced by the ‘Green Revolution’ [24]. Studies conducted by Noleppa and Cartsburg [23] indicated that plant breeding has contributed, on average for all major arable crops grown in the European Union (EU), a yield increase of about 67% since the turn of the millennium. This translates into an average yield enhancement of 1.16% per annum for the major crops. These values are higher than the individual crop yield gains reported by Evenson and Gollin [25] from 1960 to 2000. The development of high-yielding varieties with multiple disease resistances and enhanced water- and nutrient-use efficiency also has considerable societal and environmental benefits, reducing pesticide- and fertiliser-induced hazards and greenhouse gas emissions, apart from avoiding the further expansion of agricultural land [23]. In terms of production volume, similar observations have also been made for tomatoes, the globally dominant vegetable crop, and alfalfa, a globally important forage crop [26].

Breeding and agricultural intensification efforts led to a significant availability of food, which, in turn, contributed to a notable decline in the number of people suffering from chronic hunger. However, after years of steady decline, the trend in world hunger reverted in 2015 and remained relatively constant until 2019 (618.4 million undernourished; 8%). From 2019 to 2020, the prevalence of undernourished people rose sharply, from 8.0 to 9.3%, and to 9.8% in 2021, meaning that approximately 767.9 million people were affected by hunger in 2021 [27]. Current projections indicate that close to 670 million people, or about 8% of the global population, will still face chronic hunger in 2030, approximately the same proportion of the population as in 2015 when the Zero Hunger target of the 2030 Agenda for Sustainable Development was launched by the United Nations [28]. In 2021, 425 million people in Asia, 278 in Africa and 56.5 in Latin America and the Caribbean were suffering from hunger. All in all, around 2.3 billion people (nearly one-third of the world population) were moderately or severely food insecure in 2021 and suffered from chronic micronutrient deficiencies [27,29].

Promoting the production and consumption of vegetables (and fruit) is a valid approach to alleviating ‘hidden hunger’ and enhancing nutrition security, especially in the case of diets that are dominated by high-energy foods with low levels of micronutrients [30]. This requires significant efforts in crop breeding for sustainable intensification and adaptation to changing climates. During a recent 10-year period (2008–2018), there was indeed a significant increase (24%) in global commercial vegetable production, mainly attributable to production increases in Africa (32%) and Asia (28.3%) [31]. 

According to the Food and Agriculture Organisation, global vegetable and fruit production in 2020 was estimated to be around 1128 and 887 million metric tons, respectively [32], which would result, in theory, in vegetable and fruit availability of almost 700 g per person per day, assuming 8 billion consumers. This amount is well above the 400 g of fruit and vegetables recommended for daily consumption by the World Health Organisation (WHO) [33] but does not reflect the much lower edible portions of the harvested produce and considerable losses along the value chain. By 2015, only 55% of the global population had an average fruit and vegetable availability above WHO’s minimum intake target (400 g), while people in Sub-Saharan Africa, on average, only have access to about 200 g of fruits and vegetables per day [34].

Crop domestication and improvement were based on intentional, ongoing selection for traits that improved the quality and palatability of plant organs for human consumption, facilitated crop cultivation and harvesting (e.g., suitable for mechanical harvesting and non-shattering seeds), enhanced yield and productivity, resistance against pests and diseases and tolerance to a variety of environmental stresses [35,36]. Professional plant breeding basically started with the re-discovery of the laws of inheritance by Gregor Mendel, first published in 1866 in the Proceedings of the Natural History Society of Brno, 157 years ago [37]. Many scientists consider Mendel the father of modern genetics. Various methods are used in plant breeding [38,39]. They can be based on the visual selection of plants with desired variants occurring in nature or within traditional varieties. Often, new genetic diversity is introduced into breeding populations by intercrossing selected elite plants with desired traits that complement each other or by introgression of desired traits/genes from CWR into an advanced breeding line. Modern marker-assisted precision breeding is based on monitored recombination of specific genes with the help of molecular tools that systematically track within-genome variation. 

The choice of the breeding method being applied is often crop-specific, determined by the mode of reproduction and the breeding objectives [39]. In the commercial breeding of vegetable crops, the production of hybrids is steadily increasing, as it allows the exploitation of heterosis and facilitates the multiple stacking of desired traits. Careful pollination control is required to ensure efficient hybrid production. Depending on the crop, technologies that inhibit pollen production in mother plants may include manual or mechanical emasculation and genetically controlled systems, such as male sterility [40]. Once desired traits have been fixed in a new variety, and genetic uniformity, yield stability and local adaptation have been verified, seed production and commercialisation of the new variety commence.

## 3. Current and Evolving Policies and Procedures for Germplasm Collecting, Conservation, Exchange, Use and Related Benefit-Sharing—CBD, International Treaty, Nagoya Protocol, DSI Debate

The intergovernmental negotiations that led to the signing of the CBD in 1992–1993 made member states agree to conserve the biological resources existing in their respective territories, make them available and share benefits deriving from their use on agreed terms. These terms were further specified and formalised in the Nagoya Protocol, which entered into force in October 2014 [41]. With the objective of harmonising the existing access and benefit-sharing (ABS) regulations for plant genetic resources for food and agriculture (PGRFA), as established by the FAO Commission for Genetic Resources for Food and Agriculture (known as International Undertaking), with those established under the CBD, the International Treaty for PGRFA (ITPGRFA) was negotiated by the member states of FAO, adopted in 2001 and came into force in 2004. As part of the ITPGRFA, the Multilateral System (MLS) addresses facilitated access to PGRFA for specific uses, such as conservation, research, breeding and training for food and agriculture. The MLS provides a transparent, multilateral access and benefit-sharing mechanism for both providers and users of PGRFA through the signing of a Standard Material Transfer Agreement (SMTA), thus reducing transaction costs. However, it only covers a limited number of so-called Annex I crops [42] but includes the collections maintained by the CGIAR and other international agricultural research centres that fall under Art. 15 (see below). For access to germplasm of other crops and plant species not covered by the MLS and Art. 15 of the ITPGRFA, the ABS regulations of the CBD, as specified under the Nagoya Protocol, are applicable.

Navigating and complying with these international treaties and their specific ABS regulations, especially the Nagoya Protocol, is complex. There is apparent ambiguity about the scope and application of many provisions of the international agreements regulating access to PGRFA and related benefit-sharing schedules. Access to vegetable genetic resources is severely restricted, especially since most vegetable crops are not included in the list of Annex I crops of the ITPGRFA. Access to genetic sequence information is still being debated and is subject to change.

### 3.1. The Convention on Biological Diversity

Prior to the intergovernmental agreement of the CBD, plant genetic resources were recognised as a common heritage of humankind [1,2] and could be, as stipulated in the IU established by FAO in 1983, freely collected, integrated into and accessed from genebanks, and shared with other countries and public and private genebanks [3,43]. This approach changed completely with the CBD entering into force in December 1993, which had as its main objectives the conservation of biological diversity, the sustainable use of its components and the fair and equitable sharing of the benefits arising from the utilisation of genetic resources [44]. Under the CBD, states can use their national sovereignty to exercise control over the genetic resources within their national territories. According to Article 2 of the CBD, genetic resources “mean genetic material of actual or potential value”, and genetic material is defined as “any material of plant, animal, microbial or other origin containing functional units of heredity” [45]. Thus, the CBD covers a broad range of wild and cultivated biodiversity occurring and originating in the 196 member countries and any current and potential use of this diversity [13]. Human genetic resources and genetic resources occurring beyond national jurisdictions are excluded from CBD coverage. 

Access to genetic resources is granted under prior informed consent (PIC) and mutually agreed terms (MAT) by the party owning and providing those resources (Article 15 of the CBD) [46]. The CBD does not define or describe terms of ABS of genetic resources but expects the contracting parties to develop and implement national laws to that effect (Articles 6 and 15), based on guidelines provided in the Nagoya Protocol.

### 3.2. The Nagoya Protocol

The Nagoya Protocol on Access to Genetic Resources and the Fair and Equitable Sharing of Benefits Arising from their Utilisation is a supplementary agreement to the CBD [41]. It was adopted in October 2010 in Nagoya, Japan, and entered into force in October 2014. The Nagoya Protocol focuses on the third objective of the CBD, the fair and equitable sharing of benefits arising from the utilisation of genetic resources. It aims to create greater legal certainty and transparency for providers and users of genetic resources by establishing predictable conditions for access to genetic resources and helping to ensure benefit-sharing when genetic resources leave the provider country. The Nagoya Protocol has a membership of 140 Parties (140 ratifications and 92 signatories) (https://www.cbd.int/abs/nagoya-protocol/signatories/; accessed on 31 July 2023). Contracting parties of the Nagoya Protocol are required to implement and adhere to measures regarding access to genetic resources, benefit-sharing and compliance. Provider countries must set out clear conditions for granting PIC and establishing MAT, including access to traditional knowledge associated with genetic resources that are held by indigenous communities, while user countries are required to monitor use and compliance with the MAT. 

The Nagoya Protocol refers to several tools and mechanisms that assist with implementing its objective. Among those are the establishment of national focal points (NFPs) and competent national authorities (CNAs) that serve as contact points for information, grant access and cooperate on issues of compliance [41]. Other provisions include an Access and Benefit-sharing Clearing-House (ABSCH) to share relevant and updated information on national regulatory ABS requirements and contact details of NFPs and CNAs [47]. Unfortunately, the information available on the ABSCH website is incomplete and, for a number of countries, not up to date. It does not provide all the details required to access PGR in a particular country. Addresses of NFPs are often not indicated, the CNAs are not clearly defined and clear ABS instructions are often missing. Moreover, law texts are often only available in the local language. 

The text of the Nagoya Protocol also provides an Annex with a list of possible monetary and non-monetary benefits that can satisfy the Convention’s requirement for benefit sharing (https://www.cbd.int/abs/text/articles/?sec=abs-37; accessed on 31 July 2023). The non-monetary benefits listed under the Nagoya Protocol are numerous and include, among others, sharing of know-how, collaboration, cooperation and contribution to scientific research and development programmes. The value of those non-monetary benefits is often overlooked in the current heated discussions, which mostly centre around monetary benefits. The social, economic and ecological benefits and impacts that accrue to countries through the sharing and use of improved and well-adapted crop germplasm and associated technologies might far exceed the levels of monetary benefits that could be generated under the benefit-sharing arrangements of the CBD and the ITPGRFA [13,48]. 

### 3.3. The International Treaty for Plant Genetic Resources for Food and Agriculture

The International Undertaking (IU) on Plant Genetic Resources, a voluntary agreement established by the FAO Commission on Genetic Resources for Food and Agriculture in 1983 to promote international collecting, conservation, exchange and use of plant genetic resources for breeding and scientific purposes [2], conflicted with the ABS policy of the CBD, allowing each contracting party to exercise sovereign rights over natural resources. As a consequence, the IU became obsolete and the FAO started its revision process in 1994. The negotiations of the FAO member states were concluded in November 2001 with the adoption of the International Treaty, which came into force in June 2004 [49]. The ITPGRFA is in full compliance with the CBD. In contrast to the bilateral negotiations required under the CBD/Nagoya Protocol, the ITPGRFA establishes, through its MLS, a globally harmonised system to provide farmers, plant breeders and the scientific community with transparent and facilitated access to plant genetic resources. The International Treaty also ensures recipients share the benefits of using these genetic resources. As of 1 January 2023, the ITPGRFA has 150 Contracting Parties, including the European Union as a member organisation (https://www.fao.org/plant-treaty/countries/membership/en/; accessed on 31 July 2023).

An essential provision of the ITPGRFA is the multilateral system (MLS) that facilitates access and benefit-sharing of PGR of crops specified on a defined list, known as Annex I, consisting of 35 food and 29 forage crops. From the contracting parties, all genetic resources that are in the public domain and under governmental control and management should be included in the MLS and as such made freely available (or at minimal transaction cost) for the purposes of conservation, research, breeding and training under a Standard Material Transfer Agreement (SMTA), instead of the use of prior informed consent and mutually agreed terms, on a country-by-country or case-by-case basis, as prescribed by the CBD. However, if the PGRFA materials in the MLS are intended for other purposes only, such as biofuels, pharmaceuticals or other industrial uses, they are excluded from the scope of the International Treaty’s MLS [13]. The germplasm recipients shall not claim any intellectual property or other rights that limit access to these resources or their genetic parts or components in the form received from the MLS [49]. 

As outlined in Article 15 of the ITPGRFA, the MLS also includes Annex I PGRFA materials that are maintained by genebanks of the CGIAR or other regional or international agricultural research centres that have signed agreements with the Governing Body of the International Treaty [42]. These agreements with the Governing Body stipulate that the international centres will also make non-Annex I plant materials held in their collections, under their 1994 “in trust” agreements with the FAO, available to users under the same SMTA terms as foreseen for the exchange of Annex I germplasm [13]. As of August 2018, the genebanks of the CGIAR centres conserved and made more than 730,000 accessions of crop, tree and forage germplasm available under the MLS [48]. 

The European Cooperative Programme for Plant Genetic Resources (ECPGR) stated in its 2016 report under Article 32–ECPGR and the Treaty/Nagoya Protocol: “It is recommended that all ECPGR member countries (36 as of June 2023 for the current phase (2019–2023)) as appropriate and in line with national legislation, use the SMTA for distribution of both Annex I and non-Annex I PGRFA accessions, independently of whether material is conserved in ex situ collections or held in situ” [50]. Meanwhile, a few countries, such as Germany and the Netherlands as well as NordGen, acting for the Nordic countries (Denmark, Finland, Iceland, Norway, Sweden and the autonomous territories of the Faroe Islands and Greenland) [51], have already adopted this recommendation. The Nordic countries signed the so-called Kalmar Declaration, which recommended facilitated access to PGRFA in general. Greenland is the only Nordic country that introduced “classic” ABS legislation with requirements for PIC and MAT for access to genetic resources or related TK, while Norway has enacted an authorisation for such legislation concerning wild genetic resources. For access to TK held by indigenous peoples and local communities (ILCs), Norway has enacted legislation that requires PIC for the use of TK associated with genetic resources from ILCs. Finland has enacted a PIC requirement for using TK held by the Sámi people, and Sweden is establishing formal consultation with the Sámi Parliament of Sweden on matters dealing with TK [51].

The large collection of plant genetic resources of both Annex I and non-Annex I material of the NordGen genebank has been placed in the Nordic public domain, making all accessions available through the International Treaty’s SMTA. It is interesting to note that from 2018 to 2020, NordGen had to deal with a considerable increase (89%) in the number of ordered seed samples, primarily requested by Nordic and other European countries [51]. This could be attributed to the increased complexity of obtaining germplasm from elsewhere. It is also important to state that all accessions (including non-Annex I) designated by ECPGR member countries to the European collection AEGIS (A European Genebank Integrated System) are distributed to users under the terms of the SMTA (https://www.ecpgr.cgiar.org/aegis/about-aegis/aegis-and-the-treaty; accessed on 31 July 2023). 

According to Article 13 of the ITPGRFA, facilitated access to germplasm itself is considered to be an important benefit. Other non-monetary benefits include the exchange of information, access to and transfer of technology and capacity building [42]. In practice, this happens to some extent in public–private partnerships between genebanks and the private breeding sector through multiplication and/or evaluation of selected accessions. The SMTA of the ITPGRFA also foresees monetary payments by germplasm recipients under Article 6.7, if material received under an SMTA is used to create PGRFA materials that are not freely available for research and breeding by others (i.e., 0.77% of the sales of those PGRFA or, according to Article 6.11, an alternative payment of 0.5% of all sales of PGRFA belonging to the same Annex I species, to an international benefit-sharing fund (www.fao.org/plant-treaty/areas-of-work/benefit-sharing-fund; accessed on 31 July 2023)). This fund is used to support the conservation and sustainable utilisation of PGRFA, especially among local traditional farming communities. 

Payments under SMTAs take time to materialise, as product development with the received germplasm and testing the final product (variety) containing this germplasm is time-consuming. In addition, it is important to consider that most breeders may not seek Intellectual Property protection through patent application for new varieties, hence not limiting the free use of the final product (variety) for further breeding and thus not resulting in mandatory payments. In 2013, the Governing Body of the ITPGRFA established an intergovernmental Open-ended Working Group to negotiate and recommend measures to enhance the functioning of the MLS under the ITPGRFA for consideration by the Governing Body. Various stakeholder groups (seed industry, CGIAR, farmer organisations and civil society) contributed to this process [52]. A group of seed companies proposed a ‘subscription-only model’, which was meant to replace one-time payments for receiving and using a specific genetic resource in case the final product is not freely available for further breeding. Companies would instead contribute annually a percentage of their annual seed sales for selected Annex I crops as listed under the ITPGRFA for an initial 10-year period, thus ensuring an ongoing and consistent contribution to the ITPGRFA’s Benefit-sharing Fund [53] from the very moment of signing the subscription agreement for one or more selected Annex I crops. Forty-one members of the International Seed Federation (ISF) indicated that, if the subscription covers all crops, a single rate of 0.01% on sales of Annex I crops would be acceptable [54]. For those companies that would not participate in the subscription system, the continuation of a single access mechanism with payments based on the use of accessed genetic resources as reflected in current Articles 6.7 and 6.8 of the SMTA [54] should remain, next to the subscription model. ISF indicated that its members would be ready to make payments based on Articles 6.7 and 6.8, as long as the Article 6.8 option was manyfold lower than the payment under Article 6.7. 

The intergovernmental Open-ended Working Group considered this idea and proposed a subscription system linking payment obligations to *access* rather than *use* and commercialisation, whereby users pay a fee for being granted access to all or a selected group of PGRFA within the MLS. Such a subscription system would start from the date of signing the subscription agreement (prior to shipping any materials) and thereby would eliminate the need to track single germplasm samples obtained under the MLS down to the final commercialisation of products [52]. However, no agreement has been reached up to now because of the difference in opinion on payment rates, DSI and the expansion of Annex I to all PGRFA. The negotiations were stopped during the eighth Session of the Governing Body of the ITPGRFA (GB 8) in November 2019. 

During GB9 held in September 2022 in New Delhi, India, delegates agreed to re-establish the Ad Hoc Open-ended Working Group to Enhance the Functioning of the MLS to finalise the process of adopting a series of options/measures by GB 11 in 2026 [55]. The envisaged measures aim to (i) increase the monetary and non-monetary benefits arising from the MLS for all providers and users; (ii) ensure a sustainable and predictable long-term increase in user-based income to the Benefit-Sharing Fund (BSF); (iii) expand the MLS to include all PGRFA and improve their availability; (iv) make the MLS more dynamic and responsive to new developments and innovations, and to create legal certainty, administrative simplicity, and transparency for all participants. 

### 3.4. The Digital Sequence Information Debate

Rapid advances in genomics and the open access to digital sequence information (DSI), available to all through public sequence databases, have contributed considerably to the recent progress in plant and life sciences, including health, conservation and use of (agro)biodiversity [14]. Plant genomics is evolving rapidly; single reference genomes for plant species are now complemented by multiple sequences and pangenomes, depicting the diversity within crop species or genera.

The term DSI still lacks a precise and generally accepted definition [56,57,58]. It has been invented as a place-holder term by negotiators covering under a narrow interpretation genetic/genomic sequence data only, while under a wide interpretation, biological data, such as passport, characterisation and evaluation data, ecological adaptation and traditional knowledge may be included [59]. The increasing role and importance of DSI, the speed by which it is developed and analysed, potentially allowing the circumvention of the very use of material genetic resources, contributes to an uneasy feeling among several countries, who consider DSI as a threat to the exertion of their sovereign rights over genetic resources and associated information [59,60]. Despite the rapid progress in genomics and gene editing, DSI is not expected to replace physical access to PRGFA, but the value of DSI for research, breeding and variety development is obvious. 

The International Nucleotide Sequence Database Collaboration (INSDC) is the central foundation for global sequence information. It connects over 1700 scientific databases and platforms [61]. The INSDC provides the free core infrastructure for DSI deposition, preservation, and global dissemination as part of a scientific collaboration between the European Molecular Biology Laboratory (EMBL), an intergovernmental organisation with more than 80 independent research groups, the U.S. National Center for Biotechnology Information (NCBI), which is hosting the collaboration through its GenBank, and the DNA Databank of Japan (DDBJ) [62]. Tracking and tracing the movement of nucleotide sequence data (NSD) in GenBank, the largest public database platform, is challenging [59]. Scientists worldwide search this platform for their work; hence, any financial or administrative burden for accessing NSD will affect all scientists, globally, and limit their ability to undertake individual or collaborative research. 

There is consensus among genebank curators that open access to and free use of DSI is essential for facilitating adequate conservation and use of PGRFA [43,63]. However, contracting parties to the international treaties express contradicting views on whether and how access to DSI and benefit-sharing from its utilisation should be regulated. Countries in the Global South consider free-for-all access to sequence information (with or without the associated germplasm) to be mainly beneficial for the biotechnology industry in the Global North, and as such, counterproductive for other countries, their local communities and indigenous people who are the custodians of plant agrobiodiversity and who fear that they will not be able to benefit if access to DSI is not subject to ABS regulations of PIC and MAT under the Nagoya Protocol of the CBD [64,65].

Discussions on the current and future access to digital sequence information are ongoing in several fora, under the International Treaty, CBD, Nagoya Protocol, the multilateral Prepared Influenza Preparedness (PIP) Framework, the Antarctic Treaty (dealing with Antarctic species) and the United Nations Convention on the Law of the Sea (UNCLOS), which is developing a multilateral treaty aiming at establishing best practices that regulate access to marine genetic resources and related sequencing data, while sharing the benefits derived from such access to enhance the conservation and sustainable use of marine biological diversity in areas beyond national jurisdiction [66,67,68].

Despite the rapid progress in genomics and gene editing, the successful transfer of useful traits across different life forms appears to be rather unlikely in the near future due to the complexity of biological systems [69]. Many traits are multigenic, and gene expression may also depend on epigenetics and other factors; hence, it is not straightforward. Therefore, simply disconnecting the study and use of DNA sequences from the physical germplasm resource conserved in genebanks, from which DNA was extracted, will not suffice to solve the food and nutrition security of humankind. The combination of natural selection and professional plant breeding will still be required for crop variety development and adaptation to local agroecological conditions in the foreseeable future to ensure food and nutrition security.

Scholz et al. [14] showed that a benefit-sharing scheme for DSI modelled according to the Nagoya Protocol would likely totally disrupt plant breeding and genomics research, as sequence datasets are downloaded from INSDC 34 million times per year by 10–15 million unique users, making a bilateral system requiring permissions between an end-user and country of origin prohibitively complex. All countries provide and use DSI for basic and applied research, in both the public and private sectors. For example, DSI from Brazil is used by 111 countries worldwide, while scientists in Brazil use DSI from 153 different countries [70]. In the case of Kenya, DSI from this country is used by 79 countries, while scientists in Kenya use DSI from 83 countries. Alternative methods where DSI remains open access and benefits are shared in a fair and practical manner are mandatory and should be the aim of intergovernmental negotiators [63]. A system which requires tracking, tracing, reporting and monitoring DSI information is entirely unsustainable. Multilateral approaches, which delink access to DSI and benefit-sharing, are the most appropriate for all stakeholders involved.

## 4. Implications of the Complexity of ABS Regulations for Germplasm and Related Information for Genebank Curators and Public and Private Sector Breeders

### 4.1. Uncertainties Regarding the Current ABS Regulations

Due to the ongoing genetic erosion of valuable (agro)biodiversity, germplasm collecting and ex situ conservation are of high priority to secure the necessary genetic diversity for developing new crop varieties with resilience to biotic and abiotic stresses and of high nutritional value for the benefit of humankind [15]. Genebank curators and breeders understand the national sovereignty of countries over their genetic resources and are willing to comply with clear and realistic standardised ABS regulations established by the countries in which collecting germplasm material is of high priority. The private sector also encourages clear, transparent and easy-to-implement ABS agreements for the sustainable use of genetic resources [71]. Germplasm users and conservationists are ready to share benefits, especially through capacity development in the countries where the collecting is taking place. However, many germplasm collectors struggle with the complexity and lack of clarity of how this sovereignty is exercised and interpreted by individual countries [12,43,58]. The same is true for many breeders, as clearly described by Michiels et al. [10]. Seed companies need to keep themselves up to date with the complexity created by up to 200 different national ABS frameworks, which keep changing and evolving over time. Moreover, there are further differences at provincial or local levels within a given country. Uncertainties regarding ABS regulations and procedures to secure ABS compliance include the questions on how and with whom ABS can be negotiated bilaterally, who is subject to those conditions, how compliance is monitored and how regulations apply to plant biodiversity beyond the time frame of the current instruments [10,58]. 

Only 35 food and 29 forage crops are covered by the MLS of the ITPGRFA and can be relatively easily accessed under the terms of the SMTA, provided they are placed under the MLS. Access to other food and particularly vegetable crops, including a wide range of underutilised genera with current or potential value as food plants, is governed instead by CBD/Nagoya Protocol-based diverse national regulations, and each single resource exchange needs to be negotiated on a case-by-case basis [72]. 

### 4.2. The Issue of Stacking Obligations, Retroactive Effect, Tracking and Tracing and Related Costs

The complex, and often unclear ABS regulations, implemented with a high degree of variation among countries, sometimes with retroactive effect, and the use of different germplasm sources in the long process of developing new varieties require extensive tracking and tracing and raise associated costs and administrative burdens for genebanks, botanic gardens and public and private breeders alike [10,12]. A typical example of a retroactive effect is the case of a melon variety, accessed by an American seed company from the USDA genebank, which originated in and had been received from India long before the country introduced its Biological Diversity Act in 2002. The seed company derived progenies resistant to Cucurbit Yellow Stunting Disorder Virus (CYSDV) from this melon variety and received a patent on this trait from the European Patent Office (EPO). In 2016, the National Biodiversity Office filed a non-compliance case with the EPO [73]. Even the purchase of commercial seed, with unknown history, for use in further breeding in a country that does not impose ABS obligations for the use of its genetic resources may also result in non-compliance issues with retroactive effect if it has parent material in its pedigree from a country such as India [10]. Therefore, the implementation of ABS laws in one country may result in legal uncertainty at the global level.

Moreover, the development of improved, resilient varieties may stack obligations and costs as it involves the incorporation of PGRFA, traditional knowledge (TK) and/or DSI from several countries. Every cross made by plant breeders stacks the contractual ABS requirements of its parental lines. Once a plant breeder has decided to make a cross, i.e., has ‘utilised’ a genetic resource in a cross, this resource is part of the genotype of all progenies. A sudden request to stop using a specific genetic resource in an ongoing breeding process due to the entry into force of new ABS regulations or different interpretations of their scope requires discarding all breeding materials that used the specific genetic resource in question.

The complexity and legal uncertainty regarding access to in situ and ex situ genetic resources and traditional knowledge (TK) in provider countries are described in great detail for selected countries by Michiels et al. [10]. The legal uncertainty ensuing from the complex and unclear national ABS regulations and the length of time needed and the costs involved in navigating the ABS regulations and negotiating PIC and MAT will often obstruct rather than facilitate ex situ conservation and associated research by genebanks and research institutions [12] as well as R&D investments by the public and private sector into horticultural innovations [10,74]. The efforts in terms of time and human resources that seed companies need to spend on ABS tracking and compliance issues are often considered disproportionate to the benefits that are generated and shared through bilateral ABS agreements [10]. According to Rabitz [75], costs related to germplasm access and use arise from three distinct sources: (a) transaction costs associated with access (potential administrative barriers, negotiation of bilateral ABS contracts, involvement of lawyers for clarification of uncertainties regarding legal and regulatory requirements); (b) the costs due to benefit-sharing obligations, including the costs of mandatory or voluntary tracking of PGRFA through the value chain; and (c) compliance costs after accessing the genetic resource, including monitoring costs to follow the utilisation of accessed materials throughout the value chain and to provide documentary evidence of utilisation in accordance with the applicable ABS laws and regulations. The described costs are considerable and might serve as a disincentive to access new genetic diversity. Therefore, tracking and monitoring mechanisms should be low-cost and should exclude standard genetic or biochemical analysis of individual samples since the cost of such analyses might exceed the expected benefit-sharing levels [11]. Rather than risking a company’s reputation by trying to comply with unclear national ABS regulations, which could potentially result in a non-compliance case, companies might prefer to stay away from accessing such genetic resources. Complex ABS rules may also serve as a disincentive for the development of public and private breeding programs for more regional, less profitable crops, thus even threatening progress towards CBD objectives and the United Nations’ SDGs. 

The mentioned uncertainties also adversely affect collaboration among genebanks as well as the collaboration between genebanks and breeding companies. Genebank curators will be more hesitant to rationalise their own collections by reducing duplication with other genebanks since they cannot be sure of access to other collections in the future [43]. Such uncertainties are forcing countries to stockpile plant genetic resources to ensure future access to genetic diversity for their own research organisations and plant breeders, resulting in redundancies and further stress on the already limited capacity of the PGRFA community. Similarly, private-sector breeding companies feel compelled to stockpile and conserve for the long term the currently available PGRFA in working collections and breeding lines in already existing or yet-to-be-established private genebanks.

### 4.3. The Added Complexity Due to DSI Inclusion

The current stalemate regarding the inclusion of DSI into existing ABS mechanisms or the creation of separate ones for DSI only is of major concern. Genebank curators, conservationists and plant breeders agree that access to and use of DSI is essential for adequate conservation, sustainable use of plant genetic resources [43] and the development of elite crop varieties. A major headache for genebank curators is the unresolved definition of DSI, as genebanks share germplasm with associated accession-level information [59]. Should countries pre-emptively include DSI in their national ABS legislation before standard access mechanisms have been agreed upon in international fora, genebanks may no longer wish to conserve and exchange material from those countries due to the increased complexity of germplasm handling and distribution and consequent compliance issues. Similarly, the botanical gardens community also fears that individual countries may implement disparate regulations regarding access to DSI as has been the case for physical access to genetic resources [57].

Dozens of countries have already adopted legislation on DSI, often obliging users to obtain PIC and/or MAT to work with DSI [59]. Among those are six African countries (Kenya, Malawi, Mozambique, Namibia, South Africa, and Uganda) that have already put DSI domestic measures of a legal, administrative or policy nature in place [76]. Cameroon has passed a new instrument on ABS according to the ABS Clearing House Mechanism of the CBD. This instrument provides that the ‘use of genetic information’ is considered an activity relating to the use of genetic resources, and as such is subject to PIC and MAT. Those examples may serve as proxies for potential future restrictions on many, or all, species of plants, thus hindering research and innovation and being counterproductive to the efforts of continuously improving and diversifying our food systems [77].

### 4.4. The Consequence of Overly Complex ABS Regulations

There is evidence that non-strategic national ABS regulations have threatened or even impeded access to genetic resources for non-commercial research [12,78,79]. In Argentina and Brazil, for example, newly implemented national ABS regulations prevented domestic research organisations from studying local biodiversity, even if such research did not involve international partners or the export of local genetic resources [80,81,82]. Unclear national ABS legislations and a high level of bureaucratisation with concomitant high transaction costs and compliance risks are also a disincentive for commercial exploration of biodiversity [83,84]. 

The abundance of diverse ABS regulations established by individual countries, based on the Nagoya Protocol, combined with legal uncertainties regarding their interpretation and implementation may hamper conservation and the exchange of biodiversity. Consequently, food and nutrition security, which are within the scope of international agreements such as the CBD, the ITPGRFA and the Sustainable Development Goals of the United Nations, may also be negatively impacted. Reduced international collaboration as a consequence of overly complex regulations for access to PGRFA and associated information will likely slow down capacity building and technology transfer to less advanced countries, thus deepening global inequalities. 

## 5. Options for Addressing Current Constraints on Access and Benefit-Sharing of Genetic Resources and Related Information, at the Policy Level

### 5.1. Expanding the MLS of the ITPGRFA to Cover All PGRFA, Combined with a Subscription System

Most plant breeders would prefer the open-source option of the heritage of humankind principle as established under the IU. However, this is no longer a viable option. The MLS of the ITPGRFA significantly reduces the burden of tracking and tracing and associated costs, avoids bilateral contracts and is, therefore, an option widely supported by public and private-sector breeders. In the current situation, the most straightforward ABS option for PGRFA would be an expansion of the MLS of the ITPGRFA to cover all genetic resources for food and agriculture, amongst others including all vegetable crops and species. This proposal is in line with the current efforts of the Open-ended Working Group to Enhance the Functioning of the MLS (see Section 3.3; [55]). If an expanded MLS covering all PGRFA is then based on a subscription system linking payment obligations to access rather than use or commercialisation, there would be no need to track single germplasm samples obtained under the MLS down to the final commercialisation of a new variety, plus its use by further breeding with this new variety, over and over again. Alternatively, if users do not want to adopt the subscription system, commercial utilisation of germplasm obtained under the MLS could be determined with the help of intellectual property tools, aided by using digital object identifiers (DOIs) [85]. Confirmed commercial utilisation of genetic resources would then trigger monetary benefit-sharing payments [56].

### 5.2. MLS of the ITPGRFA Covering all PGRFA, Combined with a Seed Sale Tax or Levy Option at the National Level

A seed sales tax or levy paid by contracting parties of the ITPGRFA would allow easy access to PGRFA under an expanded MLS covering all PGRFA, without the need for tracking and tracing measures to follow the movement of PGRFA up to the final product: a newly released variety. For 12 consecutive years, Norway has paid annual contributions to the benefit-sharing fund of the International Treaty, equivalent to 0.1% of the value of annual seed and plant material sales in the agricultural sector in Norway [86]. The CGIAR is recommending that contracting parties make annual payments to the Plant Treaty’s benefit-sharing fund based on seed sales within their jurisdictions (similar to the Norway levy), using a fixed royalty rate that corresponds to the value of access to, and use of, both PGRFA and DSI. Contracting parties would then have the option to recoup a portion of that levy payment from commercial users in their jurisdictions [63]. Other countries could also follow the example of Germany, the Netherlands and the Nordic countries and share all PGRFA, whether they are within or outside of the MLS of the ITPGRFA, under the SMTA. 

### 5.3. Harmonised, Multilateral ABS Regulations under the CBD/Nagoya Protocol

Harmonisation of ABS regulations under the CBD/Nagoya Protocol at the global level, in a similar manner as under the MLS of the ITPGRFA, would genuinely facilitate the work of all curators of collections at genebanks and botanic gardens and users in the public and private sectors alike [10]. The European Union’s ABS Regulation No. 511/2014 (https://eur-lex.europa.eu/legal-content/EN/TXT/PDF/?uri=CELEX:32014R0511; accessed on 31 July 2023) is a first step in that direction. However, this directive defines only the compliance measures that must be followed by all 27 member states of the EU to secure ABS compliance under the Nagoya Protocol, while ABS regulations still follow the principle of national sovereignty and can vary significantly from country to country. 

### 5.4. Harmonised National ABS Regulations and Compliance Measures

There is an urgent need to improve the current bilateral ABS systems. To effectively guide germplasm users trying to comply with all the different national ABS rules and regulations under the Nagoya Protocol, it would be conducive if the country profile on the ABS Clearing House website contained factual, concise information in English on how to access PGRFA in a given provider country and with whom to negotiate PIC and MAT, also including TK. All national legislation should be published on the ABS Clearing House website uniformly, including the scope and jurisdiction of the applicable ABS regulations and measures. A summary of the steps and obligations to follow should also be included. 

### 5.5. Options for the Contentious DSI Issue

The term DSI is still used as a placeholder term and lacks a generally accepted definition. For most scientists, genebank curators and breeders, a narrow interpretation restricting DSI to sequence information only would be preferred, as other information associated with germplasm (passport, characterisation, and evaluation data) could be shared with users together with the exchange of germplasm as is still common practice. In case of a wider definition of DSI, Lawson et al. [66] proposed two options consisting of (i) a risk framework matrix for valuing information as part of the ABS transaction by attributing an estimated value to a particular kind of information; alternatively, (ii) a charge, tax or levy that would externalise the costs so that information would remain available and accessible to all, thus benefitting the global scientific community. Under the matrix option, passport data on accessions would be considered of low value, available without restrictions (public domain data), while descriptive (phenotypic) data would be treated as restricted public access data, and sequence data would have an embargo period until the results obtained need to be reported to the germplasm/DNA provider. Under the tax or levy model, the party accessing the resources would need to pay a tax (called ‘Partnership Contribution’ under the PIP Framework) or it would be levied on contracting parties, similar to the Norway seed sales tax under the MLS of the International Treaty. The tax or levy option is appealing, as it avoids the high transaction costs required to negotiate the value of information in every single transaction and allows the scientific community to disclose and share the generated information freely [66].

Given the complexity of the DSI issue and the diversity of stakeholders involved, Scholz et al. [61] proposed the creation of a public–private partnership (PPP) to govern the implementation of any future policy framework around DSI and offered five policy options for the sharing of monetary benefits. 

### 5.6. Summary and Concluding Remarks of Section 5

Curators of plant genetic resources and public and private sector breeders hope for a transparent, functional and expanded multilateral system under the International Treaty covering all PGRFA, thereby erasing all legal uncertainties and minimising transaction costs for conservers and users of genetic resources and DSI. The authors of this paper strongly support a single, multilateral access mechanism for both PGRFA and DSI, if necessary combined with a subscription system as currently being negotiated under the MLS of the ITPGRFA or with a national tax or levy, similar to the Norwegian seed sales tax. If current and future international, regional, national and bilateral collaborative efforts would be guided by a focus on the promotion of inclusive innovation and enhanced equity in research, utilisation, and commercialisation of (agro)biodiversity and broader public and social benefits from the outcomes of science, instead of a predominant focus on immediate monetary benefits, greater benefits for all could be expected over time. 

## 6. Recommendations and Concluding Remarks

Breeding improved varieties is a continuous and even cyclic effort that is essential for enhancing food and nutrition security. Crop improvement depends on access to biodiversity to source new genetic variation for breeding. Fair and non-bureaucratic rules to access and use germplasm in breeding is therefore a predisposition for food and nutrition security. Providers and users of plant genetic resources need clear information on the conditions under which the germplasm material can be accessed and used for research and breeding. It has to be clear whether the ITPGRFA, the CBD/Nagoya Protocol or any other ABS tool applies. Furthermore, adjustments to the current texts of these legal instruments are clearly needed to ensure legal certainty and strengthen access to genetic resources. Extending the list of Annex I crops of the ITPGRFA to include all PGRFA, as well as related organisms like pathogens and pests, would greatly benefit the use of new germplasm in breeding and lead to the creation of improved varieties that can cope with climate change challenges and will contribute to more sustainable forms of agriculture. Identification and documentation of the flow of benefits from the use of plant genetic resources to the different stakeholders could contribute to a better understanding of the value of plant genetic resources and related research on this material for humankind. Such a move might reduce current tension between germplasm providers and users, and eventually lead to more transparent and easy-to-follow access provisions. Crop diversity can only benefit humanity if it is not only conserved but also used.

Germplasm conserved in genebanks is most useful when it is distributed together with relevant information. Clarity on the scope of biodiversity data subject to ABS is essential for any future progress. High-throughput approaches have greatly improved genotypic and phenotypic data collection from genebank accessions. Such information can be used to strengthen germplasm management, elucidate questions regarding the taxonomy of accessions, assist in germplasm exchange through diagnostic tools for the detection of viruses and other pathogens, as well as for selecting plant genetic resources and specific traits for research and breeding. Such information can also assist in determining gaps in existing collections and help fine-tune the objectives of new collecting missions. These data could also be used to train artificial intelligence (AI) tools for a wide range of purposes, including ecophysiological crop modelling and identifying germplasm material adapted to climate change. 

The outcome of the debate on the nature of DSI and the conditions for access and its use will be critical to actually using the data generated for plant genetic resources in research and breeding. Bilateral provider–user interactions for the use of DSI may be far too complex for regulating the DSI information flow. DSI policies should acknowledge the importance of using DSI across low-, middle- and high-income countries and strive to preserve open access to this crucial common good [14]. Non-monetary benefits that help bridge the scientific and technological gaps in developing countries should also be considered, as these stimulate international public–private partnerships and collaborations [81]. Such non-monetary benefits should include capacity building and technology transfer.

Curators of plant genetic resources in genebanks and botanical gardens as well as public and private sector breeders would benefit from a transparent, functional and efficient multilateral system under the International Treaty covering all PGRFA, thereby erasing all legal uncertainties and minimising transaction costs for conservers and users of genetic resources and DSI. Similarly, multilateral or fully open systems for exchanging biodiversity data are preferred by the wider scientific community [58]. The decision by Germany, the Netherlands and the Nordic countries to share all PGRFA under the ITPGRFA’s SMTA is an encouraging example.

## Figures and Tables

**Table 1 plants-12-02992-t001:** Main (legal) instruments regarding access and benefit-sharing of PGRFA and some of their main features.

Legal Instrument	Year of Entering into Force	Hosting Organisation and Location of Secretariat	Year of Termination	Main Legal Principles/Aspects	PGRFA Coverage	Number of Parties
International Undertaking (IU) *	1983	FAO, Rome, Italy	2004?	Voluntary agreement; common heritage principle	All plant species for food and agriculture	n.a.
CBD	1993	UNEP, Montreal, Canada	ongoing	National sovereignty; PIC; agreed terms for use and benefit-sharing; Cartagena Protocol (biosafety)	All plant species (focus on wild); information	196 contracting parties (31 July 2023)
ITPGRFA	2004	FAO, Rome, Italy	ongoing	MLS, SMTA	All PGRFA and information; Materials in MLS: Annex I plus Art. 15 collections (i.e., CGIAR), plus voluntarily added materials	150 contracting parties (1 January 2023)
Nagoya Protocol	2014	UNEP, Nagoya, Japan	ongoing	ABS; Clearing House Mechanism	All crops and species that do not fall under ITPGRFA	140 contracting parties (31 July 2023)

* The IU was a voluntary agreement, not a body of international law.

## Data Availability

No new data were created or analyzed in this study. Data sharing is not applicable to this article.

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
