# Peer review of "Critical Review of the Increasing Complexity of Access and Benefit-Sharing Policies of Genetic Resources for Genebank Curators and Plant Breeders–A Public and Private Sector Perspective"

_plants, 2023, doi:10.3390/plants12162992_

Round 1
Reviewer 1 Report
Thank you for this paper - it describes a very important topic. It could be improved, notably in its structure, which would, I'm afraid involve some significant work. Below my general and some more specific comments. I hope that you find the time to work on this paper again, because it deserves to be published !
1. The paper is a review rather than a research paper. The descriptive sections are strong and the literature references rich. It describes genetic resource policy development and its implications for plant breeding, agriculture and nutrition. The problem (introduction) seems to focus on policy/legal access. You add to it the prebreeding issue in section 5; the link between the two is not clearly formulated in the introduction; the first relates to (global) policies; the second more to institutional arrangements, so also conclusions/recommendations for the two issues are very different. I would advise you to focus the paper on one issue. In general: there is quite some normative language – please avoid that !!
2. The introduction is very lengthy; not all of the information provided is necessary to develop the problem statement. Also the organisation of the introduction section can be improved; there is quite some duplication or topics come back in different sections. The complexity of access is mentioned in different places – eg 173-183 and 192-2008 and 2011-2015 – the need to promote breeding in 184-192 – 209 – 211. It would be helpful to improve the line or argument in the introduction leading to a clear formulation of the problem. 226 – 233 presents the objective of the paper but it does not fit completely with the title (access/utilization) since it puts quite some emphasis on conservation as well. The words ‘ draw attention’ is not formulated as a research question and words like ‘it is critically important’ (for what?) and ‘satisfying’ are normative which should be avoided – especially in the problem definition.
3. Section 3 focuses on uncertainties which is a very important argument for users to think twice before accessing materials. The other arguments: unclarity where to go, capacity to negotiate (especially for SME-breeders), cost, . . . . are less pronounced in the structure of the paper. This could be improved in headings/organization of this section on implications. Other blocking things, such as non-harmonised national laws; stacking of rights (GR, DSI, TK) and uncertainties about future rules with possible retrospective nature contribute to uncertainty. Further – explain better why tracking is a very difficult thing, possibly in the introduction as it is part of the problem to which solutions should response.
4. Section 4 is about options. Starting with ‘the best option’ looks odd. Also here, please list and analyse options for the access and utilization issue without showing preferences up front – here you introduce benefit sharing (options and size) as the main bottleneck – that has not features in the previous three section (which are mainly about access).
5. Section 5 is about utilization – partly improving access through better documentation partly examples that link genebanks with prebreeding. It is a rather different topic after all the legal access issues.
Some specific remarks
Line 93-97 – here you make a significant change in the argument. You describe with a lot of data a success story of growth in yield – and then you use one article to blame seed laws to say that growth could have been larger. I suppose pesticide, water, innovation and genetic resource policies could have increased vegetable output as well (??)
After line 107 - start another paragraph – here you start a different line of thought
Line 116 and following: The section following 116 seems to be a brief overview of plant breeding. It is on the one hand incomplete and in some cases very detailed (gametocide chemicals). Since the paper is about genetic resources, you could limit yourself here in my view. If it is said that breeding is basically the creation of diversity and selection in that diversity – and that crossing plants, often from different origins, is a basic way to create diversity next to mutagenesis.
134 access has always been constrained by physical distance and limited knowledge. For most times in history never by legal instruments
163 the sequence of the argument seems illogical to me. Biopiracy is not possible under the heritage of mankind principle. The term biopiracy came into being after !!! the national sovereignty principle was put in international law (CBD). Biological diversity was declared a natural resource and this under the souvereignty of states – the IPR protection of products based on such biological diversity in pharma and agric has indeed likely supported the motivation.
188 – crop wild relatives – you introduced the abbreviation earlier
255 I wonder whether this is correct: the MLS only applies to genetic resources collections under the control of the government – others fall under the CBD (and Nagoya). All crops for F&A fall under the Treaty, but only some materials of these crops are in the MLS
296 139 parties and 140 signatories – I suppose the EU is also a signatory but not a party???
330 I think you are right that there was conflict between IU PGRFA and CBD, but not!! At the level of sovereign rights – these had been included in the IU in – I think 1989 , at least before 1993 when the CBD entered into force. The IU was a voluntary agreement – not a body of international law. The Treaty is!
371 – is it true that all genetic resources are shared – there are special rights for Sami people in Finland, and for the Inuit in Greenland
401 – IP-protection – you mean patent protection? PVRs do not trigger BS . . . .
484 – I think negotiations in UNCLOS were concluded in the meantime
529 ‘poorly designed’ is a very normative formulation – rephrase
598 – 605 is an essential reference for a number of claims that are made earlier – propose to move this to the introduction as it is important for the problem statement of this paper
606-613 please refer in this conclusion of the section to the title, i.e. focus here on the uncertainties rather than the outright limitations to access that the rules provide. I am not sure that uncertainty is the key issue in the use of materials for ecosystem services as introduced in this paragraph
631 rephrase the ‘should’ as ‘ harmosation would facilitate . . .
Basically, you introduce in the options not so much the access as a problem, but the benefit linked to mechanisms (and expectatios). Are there other options? Such as the Norwegian tax option, or redefinition of the benefits not – the user but the final beneficiary – the consumer, thus going for a minimal percentage of GNP? Or . . . . .
617 best option “ for breeders and genebank managers” – please include this, because countries, indigenous peoples might think differently if their focus is benegfit sharing and not conservation and access (actually, most breeders would prefer the open source option of the heritage of mankind principle)
650 DSI: her you assume that the definition of DSI I not restricted to sequence data. Would such restriction not be an important option?
679 curators . . . .hope for (do you have a reference or is this to be phrased as a conclusion that the solution that you describe is the best option – as a conclusion of the section 4????
Reviewer 2 Report
The authors have addressed a very nice and informative topic; “Critical Review of Access to and Utilisation of Genetic Resources by Plant Breeders – A Public and Private Sector Perspective”. The topic is very broad and the authors are limited to vegetables crops. Genetic resources not only include vegetables but other horticultural and agronomic crops are also part of it.
Title may be revised and limited to vegetables or other crops must be included in the literature.

Moderate editing of English language required.
Reviewer 3 Report
The authors represent a valuable review to introduce the access and utilization of plant genetic resources related to breeding. The manuscript provides a plenty of information including the international agreements and genetic resource centers with their histories.
(1) I found structure of paragraphs seem not completely being organized in a good shape. In Line 57-60, 108-112, and 163-168, one sentence composes one paragraph. One paragraph should contain at least three sentences (a topic’s sentence, at least one sentence representing contents, and a conclusion sentence).
(2) I wonder if the format of heading/subheading can be optimized or not. In line 235-237, line 514-516, and line 692-695, a long heading is presented. In line 235-237, 265, 326, and 441, parenthesis (bracket) is used.
(3) It would be better to represent Table or Figure for better readability. For example, Line 238-256 tells readers the history. It is able to be represented by a chart in parallel.
Round 2
Reviewer 1 Report
Thanks dear authors for the extensive work that has been done on the paper following the first submission. I am happy that you were able to address the several concerns that I had. This is a paper that deserves to be published!
Reviewer 2 Report
Most of the comments addressed by the authors. Manuscript is efficiently improved.
Minor editing of English language required.